# Modulation of Lipid Metabolism by Trans-Anethole in Hepatocytes

**DOI:** 10.3390/molecules25214946

**Published:** 2020-10-26

**Authors:** Ahran Song, Yoonjin Park, Boyong Kim, Seung Gwan Lee

**Affiliations:** 1Department of Integrated Biomedical and Life Science, College of Health Science, Korea University, Seoul 02841, Korea; emily3317@korea.ac.kr; 2Department of Clinical Laboratory Sciences, College of Health Science, Korea University, Seoul 02841, Korea; pyoonjin@naver.com; 3Life Together, 13 Gongdan-ro, Chuncheon-si 24232, Gangwon, Korea; 4Mitosbio, 13, Gongdan-ro, Chuncheon-si 24232, Gangwon, Korea

**Keywords:** trans-anethole, lipid oxidation, non-alcoholic fatty liver disease, 5′ AMP-activated protein kinase, cellular senescence

## Abstract

Non-alcoholic fatty liver disease is caused by excessive lipid accumulation in hepatocytes. Although trans-anethole (TAO) affects hypoglycemia and has anti-immune activity and anti-obesity effects, its role in non-alcoholic fatty liver disease remains unknown. This study aimed to evaluate the effects of TAO on cellular senescence, lipid metabolism, and reinforcement of microenvironments in HepG2 cells. To analyze the lipid metabolic activity of TAO, PCR analysis, flow-cytometry, and Oil Red O staining were performed, and mitochondrial membrane potential (MMP) and cellular senescence kits were used for assessing the suppression of cellular senescence. At 2000 μg/mL TAO, the cellular viability was approximately 99%, and cell senescence decreased dose-dependently. In the results for MMP, activity increased with concentration. The levels of lipolytic genes, *CPT2, ACADS*, and *HSL*, strongly increased over 3 days and the levels of lipogenic genes, *ACC1* and *GPAT*, were downregulated on the first day at 1000 μg/mL TAO. Consequently, it was found that TAO affects the suppression of cellular senescence, activation of lipid metabolism, and reinforcement of the microenvironment in HepG2 cells, and can be added as a useful component to functional foods to prevent fatty liver disease and cellular senescence, as well as increase the immunoactivity of the liver.

## 1. Introduction

Non-alcoholic fatty liver disease (NAFLD), the most common liver disorder, causes nonalcoholic steatohepatitis, which can further lead to severe diseases, including hepatocarcinoma, cirrhosis, and cardiovascular disease [1,2]. Metabolic dysfunctions such as excessive lipogenesis in hepatocytes, accumulated triglycerides (TGs), and cholesterol causes NAFLD [3]. 

To hydrolyze accumulated TGs, hepatocytes modulate the activation of various enzymes, including glycerol-3-phosphate acyltransferase (*GPAT*) [4], diglyceride acyltransferase [5], acetyl-coA carboxylase 1 (*ACC1*) [6], hormone sensitive lipase (*HSL*) [7], adipocyte protein 2 (*AP2*) [8], and lipoprotein lipase (*LPL*). To modulate the activation of the lipolytic enzymes, activation of AMP-activated protein kinase (*AMPK*), an upstream regulator, is conducted by hepatocytes [9]. By activating *AMPK*, β-oxidation of fatty acids is activated in hepatocytes. Notably, activated *AMPK* increases the levels of *HSL* and *LPL* associated with lipolysis. Phosphorylated Perilipin-1 (*PLIN1*), a lipid droplet-associated protein, activates *HSL* to lyse TGs [10]. In β-oxidation, acyl-CoA synthetase (*ACS*) converts the free fatty acids to acyl-CoAs and carnitine palmitoyltransferase (*CPT*) transports them into the mitochondrial matrix [11,12,13]. The converted acyl-CoAs are fragmented by acyl-CoA dehydrogenase (*ACAD*). To activate lipolysis, *ACC1*, a major enzyme in lipogenesis, is inhibited by several factors, including activated *AMPK*, glucagon, and increased levels of acyl-CoA [14]. 

Trans-anethole [TAO, 1-methoxy-4-(1-propenyl)benzene], an aromatic organic compound, is a colorless and slightly volatile liquid with a distinctive odor [15]. In nature, high levels of TAO are found in many herbs, including fennel (*Foeniculum vulgare*), anise (*Pimpinella anisum* L.), and star anise (*Illicium verum*) [16,17,18]. Likewise medicinal compounds in a plant [19], TAO is studied for its bioactive functions, including antimicrobial, antifungal, and insecticidal activity [20,21,22]. It is also known for its blood sugar lowering activity [23], and for inducing browning in white adipocytes and activating brown adipocytes [24]. Additionally, TAO suppresses tumorigenesis and oxidative stress by scavenging free radicals in humans [25]. Although there have been many studies focusing on the lipid metabolism of hepatocytes, the modulatory mechanism of TAO remains elusive. 

We aimed to evaluate the bioactive effects of TAO on cellular senescence, lipid metabolism, and reinforcement of microenvironments in HepG2 cells. We presented possibility for the utility of TAO as functional foods through the establishment of its above-mentioned bioactive effects.

## 2. Results

Under TAO, signal pathways of lipid metabolism in HepG2 cells were schematized at Figure 1. 

### 2.1. Effects of TAO on Cytotoxicity and Cellular Metabolism 

Despite high concentrations (1000, 1500, and 2000 μg/mL), TAO was nontoxic in HepG2 cells (Figure 2a). Moreover, even at 2000 μg/mL of TAO, the cellular viability was approximately 99% (Figure 2a). When exposed to 100 and 500 μg/mL of TAO, the senescence was estimated to be 0.40 and 0.37 times greater than those of the control, respectively. Although the difference in cellular senescence between the two concentrations was not significant (*p* < 0.07), senescence dramatically decreased by 0.14 times at 1000 μg/mL (Figure 2b). In the case of MMP, the activity increased by 1.50 and 1.55 times at 500 μg/mL and 1000 μg/mL of TAO, respectively (Figure 2c). As these results are for the stained cell counts, the geometric means of fluorescence intensity also had the same tendency with the counts. At 500 μg/mL and 1000 μg/mL of TAO, the fluorescence intensity increased by 1.27 and 1.59 times, respectively (Figure 2c). 

### 2.2. Analysis of the Expression Levels of Lipid Metabolism Genes

TAO activated several lipolytic genes, including *AMPK*, *ACS, ACADS, LPL, CPT1, CPT2, PLIN1*, and *HSL*, in HepG2. After exposure to TAO for one day, the levels of *AMPK*, an upstream molecule, increased by 1.35 and 2.50 times at 500 and 1000 μg/mL of TAO, respectively (Figure 3a). Under increased *AMPK* levels, the levels of *GLUT4* increased by 2.5 times in 3 days at 1000 μg/mL of TAO (Figure 3a,c). For *ACS,* a molecule downstream of *AMPK*, the levels increased by 1.59 times for 1000 μg/mL TAO (Figure 3a,c). With an increase in *CPT* level by 25.82 times, *ACAD* on day three was increased considerably by approximately 11.76 times for 1000 μg/mL TAO (Figure 3b,c). On day one, the levels of *HSL* and *LPL* were also upregulated by 1.29 and 1.45 times under the effect of 1000 μg/mL of TAO (Figure 4a,b). On day three, the *AP2* level increased by approximately 3.68 times at 1000 μg/mL TAO. *PLIN1* level increased dose-dependently and *HSL* increased considerably at 1000 μg/mL TAO (Figure 4c,d). Compared to the control, the level of *HSL* was 9.35 times higher at 1000 μg/mL TAO (Figure 4d). Correspondently with the result for SREBP1(Appendix A), in lipogenic genes, including *ACC1* and *GPAT*, TAO (1000 μg/mL TAO) downregulated the gene levels by 30% and 54%, respectively, on day one (Figure 5a,b). To validate the lipolytic effects of TAO, the lipid accumulation levels were estimated using ORO staining (Figure 5c). The number of stained granules in HepG2 cells decreased dose-dependently, and a significant TAO dose was found to be 1000 μg/mL (Figure 2b).

### 2.3. Potentiality for Reinforcement of the Microenvironment 

On day one, the levels of osteopontin (OPN) increased by 1.32 times at 500 μg/mL, and on day three, the levels of pigment epithelium-derived factor (PEDF) increased by 2.88 times. It was observed that the levels were upregulated in a dose-dependent manner (Figure 6). 

## 3. Discussion

There have been no studies, to our knowledge, investigating the effects of TAO on lipid metabolism. However, the present study demonstrated the bioactive effects of TAO in HepG2 cells. The bioactive effects of TAO are classified into three categories, including the activation of cellular respiration without cytotoxicity, the enhancement of lipolysis, and the reinforcement of the microenvironment in hepatocytes. First, TAO activated cellular respiration without cytotoxicity in HepG2 cells, even when a high concentration of TAO was used for treatment of cells. Although the cytotoxic activities of TAO have been reported, including antimicrobial and antifungal activities [20,21] and insecticidal effects [22], it showed no cytotoxicity in mouse lymphoma cells [26]. Moreover, the administration of 5000 mg/kg TAO to rabbits [27] and treatment of 3T3-L1 cells with 200 μM (29.64 μg/mL TAO) have been described previously. Despite these high concentrations (Figure 2), TAO decreased cellular senescence without cytotoxicity in the HepG2 cells. Based on these results, it is safe to say that TAO can be adequately applied to various industries, including functional foods, and medicine. As the activation of β-oxidation and inhibition of cellular senescence induces the activation of MMP, TAO enhanced the activity of MMP and inhibited cellular senescence in HepG2 cells. Activation of the mitochondrial respiratory system in hepatocytes protects against fatty liver disease by the degradation of fatty acids [28]. These results suggest that TAO is a useful compound for preventing fatty liver disease as well as maintaining cellular viability and activity.

Second, TAO strongly activated lipolysis and inhibited accumulation of TG in HepG2 cells. Lipolysis in cells is stimulated by extracellular signals, including catecholamines, glucocorticoids, and tumor necrosis factor-α [29]. Under the effect of these extracellular signals, activated *AMPK* in hepatocytes activates several metabolic pathways, including fatty acid oxidation, ketogenesis, inhibition of lipogenesis, and insulin secretion by pancreatic cells [30]. Notably, TAO at a concentration of 1000 μg/mL, upregulated the levels of *AMPK*, GLUT2 (Appendix A) and GLUT4 in HepG2 cells, on day one or three. Additionally, the activated *AMPK* inhibited SREBP1c (Appendix A) and *ACC1* and lipogenic enzymes, and activated *HSL* and lipolytic protein in HepG2 cells. TAO has a role as the extracellular lipolytic signal, including thyroxine or glucocorticoids, for hepatocytes, and modulates lipolytic signals, including *HSL*, *ACC1*, *ACS*, *CPT*1 (Appendix A), *CPT*2, and *ACAD*, in numerous lipolytic markers. In the case of obesity, increased ROS in hepatocytes induces *AP2* and macrophage-induced inflammation of hepatocytes with the help of inflammatory cytokines [31]. Based on these results, it can be concluded that TAO is very effective at suppressing NAFLD and is an important modulator for lipid metabolism in hepatocytes. Insulin resistance is an important risk factor for NAFLD, and the metabolism of carbohydrates and lipids is regulated by blood glucose levels and insulin. When there is high insulin resistance, insulin strongly suppresses *HSL* and activates lipogenesis in hepatocytes. Additionally, insulin resistance induces the reduction of GLUT4 protein in adipocytes, muscles, and hepatocytes [32]. Functions of GLUT4 include the modulation of glucose in several cell types [33], and muscle contraction and stretching [34]. Moreover, GLUT4 is a key protein for regulating glucose levels in the entire body and for the development of type 2 diabetes [35]. Additionally, approximately 80% of liver cirrhosis is induced by impaired glucose tolerance, and approximately 10% of these cases develop diabetes mellitus [35]. Upregulation of GLUT4 and GLUT2 (Appendix A) in HepG2 cells treated with TAO indicates that TAO activates the storage of imported glucose by increasing *AMPK* and ATP levels by lipolysis, besides regulating homeostasis to maintain blood sugar levels. Furthermore, based on the Appendix A, TAO activated up regulation of TG secretory genes, *ApoB100* and *ApoC3*, and down regulation of TG intake genes, *CD36* and *FABP1.* These results documented protective function of TAO for NAFLD in hepatocytes. 

Third, OPN increased by 32% on day one, and PEDF increased strongly on day three. Although OPN was overexpressed in hepatocyte injury, inflammation, liver fibrosis, and steatosis [36], optimally secreted OPN has been found to promote the hepatic recruitment of macrophages and neutrophils [37]. PEDF has various functions, including the inhibition of endothelial cell migration, angiogenesis, immunomodulatory properties, and tumorigenesis [38]. In lipid metabolism, PEDF binds to adipose TG lipase to activate lipolysis in cells. In the presence of TAO, hepatocytes enhanced the secretion of OPN, and PEDF played a role as a reinforcing factor for microenvironments, including immunoactivity, anti-apoptosis, cellular proliferation, anti-angiogenesis, and cellular survival [39] in adjacent hepatocytes.

Consequently, TAO affects the suppression of cellular senescence, modulation of lipid metabolism, and the microenvironment in hepatocytes. First, TAO enhances the respiratory metabolism by activating the mitochondrial potential and suppressing senescence in HepG2 cells. Second, TAO modulates lipid metabolism by the activation of lipolytic markers and inhibition of lipogenic markers in HepG2 cells. Third, under the effect of TAO, HepG2 cells secrete OPN and PEDF to alter processes, such as immunoactivity, anti-apoptosis, cellular proliferation, anti-angiogenesis, and cellular survival. In summary, TAO is an excellent bioactive compound that can be added as a useful component in functional foods because it is effective at preventing fatty liver disease and cellular senescence and improving immunoactivity.

## 4. Materials and Methods 

### 4.1. Cell Culture and Treatment

HepG2 cells (Korea Cell Line Bank, Seoul, Korea) were cultured in low glucose Dulbecco’s Modified Eagle’s Medium (Invitrogen, Carlsbad, CA, USA) with 10% fetal bovine serum (Sigma-Aldrich, St. Louis, MO, USA) and 100 U/mL penicillin (Invitrogen, Carlsbad, CA, USA) at 37 °C under 5% humidified CO_2_. Further, 100, 500, and 1000 μg/mL of TAO (Sigma-Aldrich, St. Louis, MO, USA), dissolved in DMSO, were prepared to treat the cultured HepG2 cells. 

### 4.2. Cytotoxicity

To determine the optimal treatment doses of TAO, cell viability was assessed by the MTT assay, using the EZ-Cytox cell viability assay kit (DAEILL LAB Service Co., Seoul, Korea). Cells were cultured in 96-well microplates and treated with various concentrations (0, 100, 200, 300, 400, 500, 100, 1500, and 2000 μg/mL) of TAO for 3 days. 

### 4.3. Oil Red O (ORO) Staining

Cultured cells exposed to various concentrations of TAO were fixed with 4% paraformaldehyde for 20 min and washed with phosphate-buffered saline. To detect neutral lipids and lipid droplets, cultured cells were stained with ORO (Sigma-Aldrich) prepared in isopropanol. Stained cells were observed under a fluorescence microscope (Eclipse Ts-2, Nikon, Shinagawa, Japan), and the intensity of ORO staining was analyzed with NIS-elements V5.11 (Nikon, Shinagawa, Japan). 

### 4.4. Conventional Polymerase Chain Reaction (PCR) for Lipid Metabolism Markers

Cultured cells in conditioned media (containing 0, 100, 500, and 1000 μg/mL of TAO) were lysed immediately with a RibospinTM kit (GeneAll Biotech, Seoul, Korea) to extract total RNA. Quantification and quality assessment of RNA was undertaken with a NanoDrop2000 spectrophotometer (Thermo Fisher Scientific, Waltham, MA, USA). Isolated RNA was converted to cDNA using CycleScript RT Premix (dT20) (Bioneer, Dajeon, Korea), and cDNA was amplified using conventional PCR (PCR PreMix, Bioneer, Dajeon, Korea) under the following conditions: denaturation steps for 15 min at 95 °C, 40 amplification cycles (95 °C for 10 s, 59 °C for 15 s, 72 °C for 30 s) for annealing, a melting curve, and a final cooling step [40]. The sequences of the PCR primers used are provided in Table 1, and the products were analyzed using iBright (FL1000, Thermo Fisher Scientific, Waltham, MA, USA) and iBright Analysis software 3.1.3 (Thermo Fisher Scientific, Waltham, MA, USA). 

### 4.5. Flow Cytometry

Fixated cells were treated with 0.02% tween 20 for 10 min and were stained with allophycocyanin-anti-*HSL* (Abcam, Cambridge, MA, USA) and fluorescein isothiocyanate-anti-perilipin-1 (Abcam, Cambridge, MA, USA). The stained cells were analyzed using a flow cytometer (BD FACScalibur) and FlowJo 10.7.0 (BD biosciences, San Diego, CA, USA) [41]. 

### 4.6. Mitochondrial Membrane Potential (MMP) and Cellular Senescence

Cells exposed to 0, 100, 500, and 1000 μg/mL of TAO were harvested immediately and their MMP and senescence were measured using a mitochondrial membrane potential assay kit (Abcam, Cambridge, MA, USA) and CellEvent™ senescence green flow cytometry Assay Kit (Thermo Fisher Scientific, Waltham, MA, USA), respectively. The treated cells were analyzed for their membrane potential using a flow cytometer (BD FACScalibur) and FlowJo 10.7.0 (BD bioscience, San Diego, CA, USA).

### 4.7. Statistical Analysis

Statistical analyses were conducted using SigmaPlot, version 12.5 software. All statistical analyses were performed using a *t*-test and one-way analysis of variance. A *p*-value < 0.05 was considered significant [40]. 

## Figures and Tables

**Figure 1 molecules-25-04946-f001:**
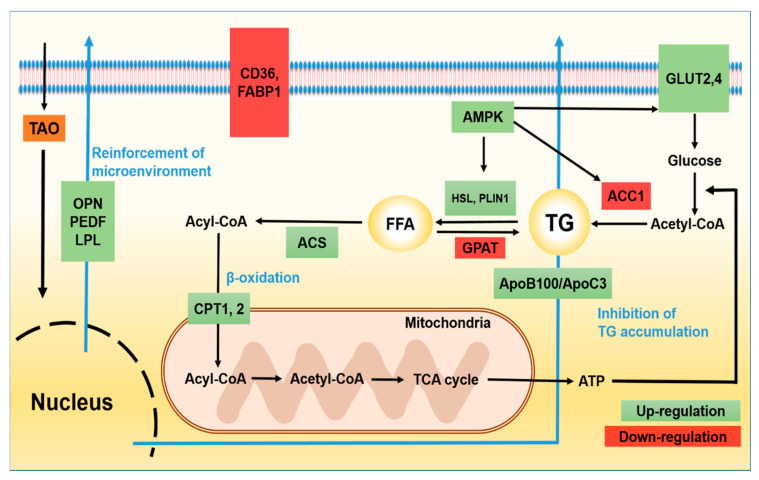
Signal pathways of lipid metabolism in hepatocytes under the effect of trans-anethole. The green boxes indicate activated signal molecules and the red boxes indicate inhibited signal molecules by TAO in HepG2 cells. TG: triglyceride, FFA: free fatty acid, *AMPK*: AMP-activated protein kinase, *HSL*: hormone sensitive lipase, *LPL*: lipoprotein lipase, *PLIN1*: Perilipin-1, *GPAT*: glycerol-3-phosphate acyltransferase, *ACS*: acyl-CoA synthetase, *ACC1*: acetyl-coA carboxylase 1, *CPT*1: carnitine palmitoyltransferase-1, *CPT*2: carnitine palmitoyltransferase-2, *ACAD*: acyl-CoA dehydrogenase, GLUT 2, 4: glucose transporter type 2, 4, ApoB100: Apolipoprotein B100, ApoC3: Apolipoprotein C3.

**Figure 2 molecules-25-04946-f002:**
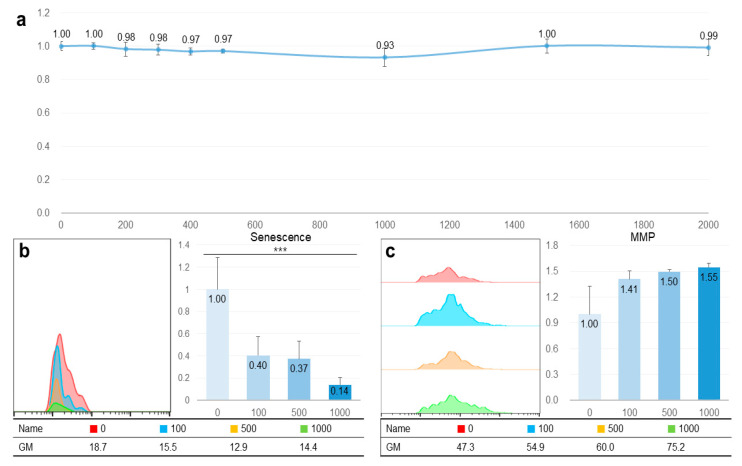
Non-cytotoxicity and catabolic activation of trans-anethole in hepatocytes. (**a**) The curve shows cellular viability in HepG2 cells under the effect of trans-anetholes. (**b**) The histogram and bar graphs show counts of senescent cells under the effect of trans-anetholes using a flow cytometer. (**c**) The histogram and bar graphs show counts for MMP activated cells under the effect of trans-anetholes using a flow cytometer. GM: geometric mean, *** *p* < 0.001.

**Figure 3 molecules-25-04946-f003:**
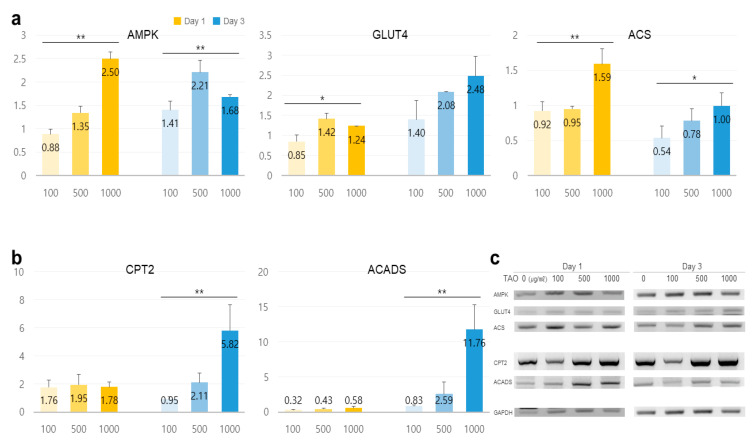
Expression of lipid metabolic markers in hepatocytes under the effect of trans-anethole. (**a**) The bar graphs show the relative fold-changes for the expression of lipolysis markers. (**b**) The bar graph shows the relative fold-changes for the expression of the β-oxidation markers (yellow: day 1, blue: day 3, * *p* < 0.05, ** *p* < 0.01.). (**c**) The gel images indicate the expression of the lipid metabolic markers.

**Figure 4 molecules-25-04946-f004:**
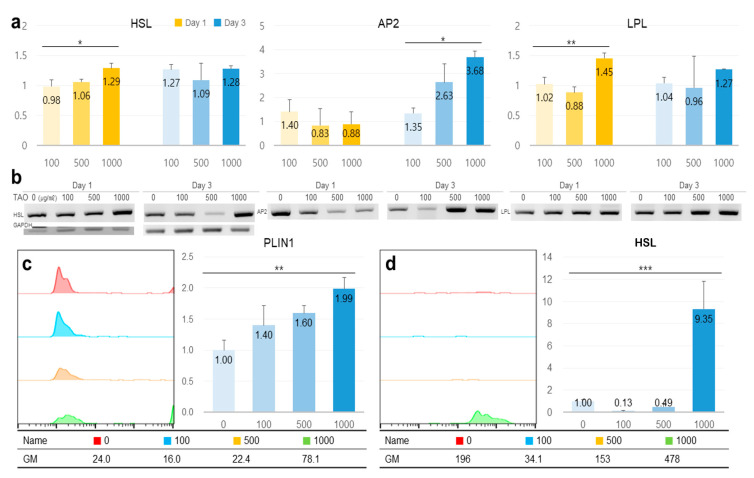
The expression of lipolytic markers in hepatocytes under the effect of trans-anethole. (**a**) The bar graphs show the relative fold-changes for the expression of lipolytic markers. (**b**) The gel images indicate the expression of the lipolytic markers. (**c**,**d**) The histograms and bar graphs show counts of perilipin-1 (*PLIN1*) and hormone-sensitive lipase (*HSL*) positive cells using flow cytometry. GM: geometric mean, * *p* < 0.05, ** *p* < 0.01, *** *p* < 0.001.

**Figure 5 molecules-25-04946-f005:**
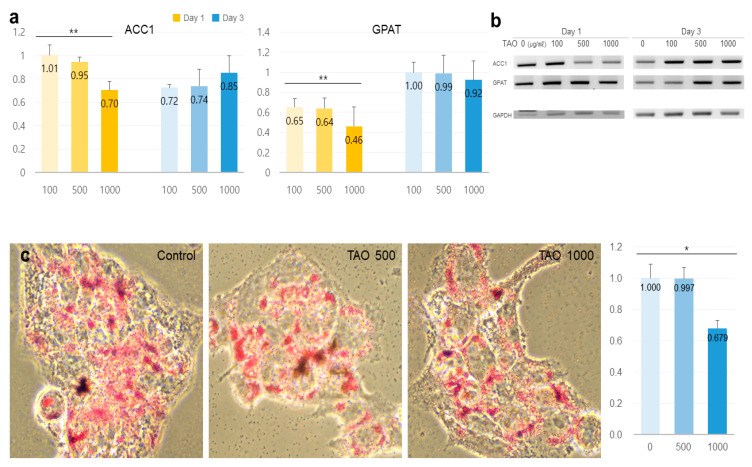
Expression of lipogenic markers and measurements of stained lipids in hepatocytes under the effect of trans-anethole. (**a**) The bar graphs show the relative fold-changes for the expression of lipogenic markers. (**b**) The gel images indicate the expression of the lipogenic markers. (**c**). The images show the Oil Red O-stained cells and the bar graphs show relative fold-changes for intensity in stained cells * *p* < 0.05, ** *p* < 0.01.

**Figure 6 molecules-25-04946-f006:**
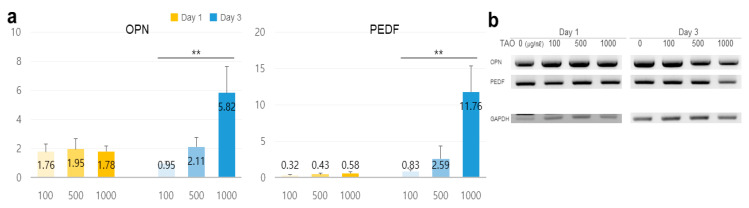
Estimation of potentiality for reinforcement of the microenvironment in hepatocytes under the effect of trans-anethole. (**a**) The bar graphs show the relative fold-changes for the expression of the reinforcers in HepG2 cells under trans-anetholes (** *p* < 0.01.). (**b**) The gel images indicate the expression of the secretory reinforcers.

**Table 1 molecules-25-04946-t001:** PCR primer sequences.

Primer	Function	F/R	Seq (5′ → 3′)	Product Size
**AMPK**	Lipid oxidation	F	CGCCTTGATTCTTTTGAGGCTT	190
		R	AGGATCAGACTACACCTGGCT	
**GLUT4**	Glucose uptake	F	TCTCCAACTGGACGAGCAAC	269
		R	AGTTATGCCACTGGTGCGTT	
**ACS**	Lipid oxidation	F	CCTGGGATCTCTCTCATGGC	289
		R	CCCCAACAACTTGCAGTGAT	
**CPT2**	Lipid oxidation	F	GACTCGGCAGTGTTCTGTCT	674
		R	GTCAGCTGGCCATGGTACTTG	
**ACADS**	Lipid oxidation	F	TTCATCAAGGAGCCGGCAAT	306
		R	AGGGTAAAGGCACATGGCTC	
**HSL**	Triglyceride lipolysis enzyme	F	AGCTGAGACACTTAGCCCCT	383
		R	CACTCCGGAGCTCTTTTTCC	
**AP2**	Lipolysis	F	TGGTGGTGGTGAGTATCTTCT	574
		R	GGTCAACGTCCCTTGGCTTA	
**LPL**	Triglyceride lipolysis enzyme	F	GGCAGCTTCATGCATTCCTC	326
		R	CAGCCAGAACGGCAACTACT	
**ACC1**	Lipid synthesis	F	GCACATCTTCACACTCCTGAA	110
		R	GTACCACTCACCTGCCGTAT	
**GPAT**	Triglyceride synthesis	F	TGGGTGAAGAATTCTGGTGGA	292
		R	CATGAGGGGTGCAGGTGTAG	
**OPN**	Changing micro-environment	F	GAATCTCCTAGCCCCACAGACC	379
		R	GTGTGAGGTGATGTCCTCGTC	
**PEDF**	Changing micro-environment	F	GCTGAGTTACGAAGGCGAAGT	102
		R	GCTGAGTTACGAAGGCGAAGT	
**GAPDH**	Housekeeping	F	GTGGTCTCCTCTGACTTCAACA	210
		R	CTCTTCCTCTTGTGCTCTTGCT	

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
