# Peer review of "Modulation of Lipid Metabolism by Trans-Anethole in Hepatocytes"

_molecules, 2020, doi:10.3390/molecules25214946_

Round 1

Reviewer 1 Report

In the present paper “Modulation of lipid metabolism by trans-anethole in hepatocytes” Song et al reported an interesting research focused on the study of the effects of trans-anethole (TAO) on cellular senescence, activation of lipid metabolism and reinforcement of the microenvironment in HepG2 cells. In particular they referred to the possible activity of this aromatic organic compound on lipid accumulation in hepatocytes by studying its contribution on the activation of lipolytic genes and on the dowregulation of lipogenic genes. They investigated the cytotoxicity of this molecule at different concentrations and they analyzed the activity of mitochondrial membrane potential after TAO treatment.

The study is stimulating but I think there are few points to be revised:

  • The authors presented the results obtained at day1 and day3 of treatment, but what about no treatment and long term treatment?
  • In the present paper the authors reported that, under the effect of TAO, HepG2 cells secrete OPN and PEDF to alter process such as immunoactivity, anti-apoptosis, cellular proliferation, etc. Maybe this point should be better investigated in order to have clear evidence regarding this aspect.

Author Response

Comment 1. The authors presented the results obtained at day1 and day3 of treatment, but what about no treatment and long term treatment?

Answer 1. Described researches, within roughly 48 hrs, fatty acid in small intestine are absorbed in vivo [1,2]. To adjust effective condition, we exposed HepG2 cells to TAO during 3 days maximally.    

[1] Kwrowska EM, Dresser GK, Deutsch L, Vachon D, Khalil W. 2003. Bioavailability of omega-3 essential fatty acids from perilla seed oil. Prostag Leukotr Ess 68:207-212

[2] Eu Deum Park et al., Absorption Evaluation of Enteric Coated Capsules Containing Omega 3 Fatty AcidsKorean J. Food & Nutr. Vol. 25. No. 4, 1027~1032 (2012),

Comment 2. In the present paper the authors reported that, under the effect of TAO, HepG2 cells secrete OPN and PEDF to alter process such as immunoactivity, anti-apoptosis, cellular proliferation, etc. Maybe this point should be better investigated in order to have clear evidence regarding this aspect.

Answer 2. We revised those sentence as the below sentence

Third, under the effect of TAO, HepG2 cells secrete OPN and PEDF to alter processes such as immunoactivity, anti-apoptosis, cellular proliferation, anti-angiogenesis, and cellular survival.

Reviewer 2 Report

Review: Modulation of lipid metabolism by trans-anethole (TAO) in hepatocytes

The authors evaluated the effect of supraphysiological levels of TAO is a hepatocyte cell culture model (HepG2) to try to understand whether this molecule can attenuate steatosis. Unfortunately, this study is severely flawed. The authors provided a very high dose of TAO (1mg/mL) to a cell line that is not steatitic without prior lipid loading (the authors did not lipid load the cells). The authors then evaluate the effects of lipid metabolism by evaluating mRNA and proteins. The authors did not perform any functional assessment of lipid oxidation, de novo lipogenesis (DNL), apoB secretion or fatty acid uptake and secretion.

The authors fail to understand that steady state intracellular lipid levels of hepatocytes is a balance of inputs (fatty acid uptake and DNL) and outputs (lipoprotein secretion, direct fatty acid secretion, beta oxidation). Understanding all five of these variables is important for understanding changes in intracellular lipid levels if observed. The authors do not show convincing evidence that TAO treatment decreases intracellular lipid content. The authors do not provide biochemical evidence of reduced intracellular TG. The experiment should also be performed on lipid loaded HepG2 cells.

The authors have a misunderstanding of fundamental hepatocyte biochemistry. The primary transporter of glucose in hepatocytes is Glut2, not Glut4. For fatty acid oxidation, CPT1, not CPT2 is rate limiting. Beta oxidative enzymes are also not typically rate limiting. If the authors believe that fatty acids oxidation is increased, please show this biochemically with a tracer experiment. Similarly, fatty acid synthase (FAS), not ACC1 is rate limiting for DNL. Did the authors evaluate SREBP1C mRNA and/or protein levels? If the authors think that DNL is down regulated, they need to use a tracer to show this. 

The authors should also evaluate whether TAO alters apoB/TG secretion if they are arguing that TAO can improve hepatic steatosis.

Other notes:

1) LPL is an extracellular lipase that cleaves TG in lipoproteins in the vasculature. This protein would not significantly contribute to intracellular TG levels.

2) AP2 is not probably relevant in isolated HepG2 cells. 

3) What is the normal physiological intracellular TAO concentration in individuals that consume high levels of fennel, anise, or supplements? How much dietary anise or supplement would be required to reach this intracellular level?

4) The manuscript states: "documented the utility of TAO for cosmetics and functional foods". This statement is not substantiated by the data presented in this study.

Author Response

Comment 1. The authors provided a very high dose of TAO (1mg/mL) to a cell line that is not steatitic without prior lipid loading (the authors did not lipid load the cells).

Answer 1. First of all, I appreciated your detail comments for our manuscript. In our research, the goals were to estimate lipid modulative functions of TAO in hepatocytes and to decide possibility of TAO as functional components to prevent fatty liver. Healing of steatohepatitis is not our goal. Based on your comments, we did additive experiments and attached the results to supplementary data or the manuscript. Although the lipid loading is important to prepare a steatitic condition, with the result of pre-staining, we decided that HepG2 cells contained adequate amount of fats to apply our experiments. In addition, compared of normal hepatocytes, hepatocarcinoma contains low concentration of β-oxidative proteins [1] and many researches for attenuation of fat [2,3,4] documented effects of extracts without lipid loading in HepG2 cells.

[1] Jacek R. WiÅ›niewski, Anna Vildhede, Agneta Norén, Per Artursson, In-depth quantitative analysis and comparison of the human hepatocyte and hepatoma cell line HepG2 proteomes, Journal of Proteomics,Volume 136,2016,Pages 234-247,ISSN 1874-3919,

[2] Forbes-Hernández, T.Y.; Giampieri, F.; Gasparrini, M.; Afrin, S.; Mazzoni, L.; Cordero, M.D.; Mezzetti, B.; Quiles, J.L.; Battino, M. Lipid Accumulation in HepG2 Cells Is Attenuated by Strawberry Extract through AMPK Activation. Nutrients 2017, 9, 621.

[3] Burdeos, G.C., Nakagawa, K., Kimura, F. et al. Tocotrienol Attenuates Triglyceride Accumulation in HepG2 Cells and F344 Rats. Lipids 47, 471–481 (2012). https://doi.org/10.1007/s11745-012-3659-0

[4] Choi, Y., Shin, H., Choi, H. et al. Uric acid induces fat accumulation via generation of endoplasmic reticulum stress and SREBP-1c activation in hepatocytes. Lab Invest 94, 1114–1125 (2014). https://doi.org/10.1038/labinvest.2014.98

Comment 2. The authors then evaluate the effects of lipid metabolism by evaluating mRNA and proteins. The authors did not perform any functional assessment of lipid oxidation, de novo lipogenesis (DNL), apoB secretion or fatty acid uptake and secretion. The authors evaluated the effect of supraphysiological levels of TAO is a hepatocyte cell culture model (HepG2) to try to understand whether this molecule can attenuate steatosis. Unfortunately, this study is severely flawed. The authors provided a very high dose of TAO (1mg/mL) to a cell line that is not steatitic without prior lipid loading (the authors did not lipid load the cells). The authors then evaluate the effects of lipid metabolism by evaluating mRNA and proteins. The authors did not perform any functional assessment of lipid oxidation, de novo lipogenesis (DNL), apoB secretion or fatty acid uptake and secretion. The authors fail to understand that steady state intracellular lipid levels of hepatocytes is a balance of inputs (fatty acid uptake and DNL) and outputs (lipoprotein secretion, direct fatty acid secretion, beta oxidation). Understanding all five of these variables is important for understanding changes in intracellular lipid levels if observed. The authors do not show convincing evidence that TAO treatment decreases intracellular lipid content. The authors do not provide biochemical evidence of reduced intracellular TG. The experiment should also be performed on lipid loaded HepG2 cells. The authors have a misunderstanding of fundamental hepatocyte biochemistry. The primary transporter of glucose in hepatocytes is Glut2, not Glut4. For fatty acid oxidation, CPT1, not CPT2 is rate limiting. Beta oxidative enzymes are also not typically rate limiting. If the authors believe that fatty acids oxidation is increased, please show this biochemically with a tracer experiment. Similarly, fatty acid synthase (FAS), not ACC1 is rate limiting for DNL. Did the authors evaluate SREBP1C mRNA and/or protein levels? If the authors think that DNL is down regulated, they need to use a tracer to show this. The authors should also evaluate whether TAO alters apoB/TG secretion if they are arguing that TAO can improve hepatic steatosis.

Answer 2.

We attached the file for supplementary results containing analytic data for the commented markers including SREBP1, CTP1, GLUT2, ApoB100/ApoC3, FABP1 and CD36. Based on the results, unlike SREBP1, FABP1 and CD36, TAO activated up regulation of CTP1, GLUT2, ApoB100/ApoC3.

Comment 3. LPL is an extracellular lipase that cleaves TG in lipoproteins in the vasculature. This protein would not significantly contribute to intracellular TG levels.

Answer 3. We revised from LPL of intracellular activity to extracellular activity in Figure1

Comment 4. AP2 is not probably relevant in isolated HepG2 cells. 

Answer 4. The reference [30] in our manuscript documented AP2 expression in hepatocytes.

[30] Lakhani, H.V.; Sharma, D.; Dodrill, M.W.; Nawab, A.; Sharma, N.; Cottrill, C.L.; Shapiro, J.I.; Sodhi, K. Phenotypic Alteration of Hepatocytes in Non-Alcoholic Fatty Liver Disease. Int J Med Sci 2018, 15, 1591-1599, doi:10.7150/ijms.27953.

Comment 5. What is the normal physiological intracellular TAO concentration in individuals that consume high levels of fennel, anise, or supplements? How much dietary anise or supplement would be required to reach this intracellular level?

Answer 5. Under the researches [1], with injecting of anethole (1–10 mg kg− 1), bioactivity was estimated in rats. With oral administering, inhibitory effect for acute inflammation was shown at 200-500mg kg− 1 of anethole [2].

[1] de Siqueira RJB, Magalhães PJC, Leal-Cardoso JH, Duarte GP, Lahlou S. Cardiovascular effects of the essential oil of Croton zehntneri leaves and its main constituents, anethole and estragole, in normotensive conscious rats. Life Sciences 78, 2365-2372 (2006).

[2] Domiciano TP, et al. Inhibitory effect of anethole in nonimmune acute inflammation. Naunyn Schmiedebergs Arch Pharmacol 386, 331-338 (2013).

Comment 6. The manuscript states: "documented the utility of TAO for cosmetics and functional foods". This statement is not substantiated by the data presented in this study.

Answer 6. We revised this sentence “ We documented the utility of TAO for cosmetics and functional foods through the establishment of its above mentioned bioactive effects.” as the below sentence.

We presented possibility for the utility of TAO as functional foods through the establishment of its above mentioned bioactive effects.

Reviewer 3 Report

Well done, indeed. I really liked Your valuable manuscript (MS). Congratulations and Bravo!

Based on its scientific merit, I can most kindly recommend Your highly informed MS for the publishing in a forthcoming issue of this rather esteemed Journal.

Generally speaking, Your intriguing MS is well written. While the text is easy and clear to read, the conclusions are consistent with the evidence and arguments presented. More precisely, they do address the main question posed as a whole.

Recommendation: Minor Revision (Minor Changes)

First of all, the English language requires some polishing. In other words, there is yet a room for the language and style improvement. Please, genially put Your efforts in such a direction.

In addition to this, the respected Authors are most affably requested to consider the citing of a single reference throughout the text of their enlightening MS related to the diversity of naturally occurring (edible plant species, known as a herbal remedy) fatty acid ingredients:

- Natural Product Research 2012, 26(8), 696–702                                            DOI: 10.1080/14786419.2010.550580                                                                                                                                                               Hopefully, this knowledgeable MS will collect a number of hetero-citations (= will be frequently cited), once when launched/published.

At least in my humble opinion, Your solid submission (MS) might be treated with a priority, if there is a space for such an option.

Taken all together, I strongly encourage the respected Authors to submit the revised form of this promising MS to Molecules, MDPI in the shortest possible time.

Last but not least, very best of (research) luck ahead to all of You.

Author Response

Comment 1. First of all, the English language requires some polishing. In other words, there is yet a room for the language and style improvement. Please, genially put Your efforts in such a direction

Answer 1. Our manuscript was trimmed by English language editing, Editage (www.editage.co.kr)

Comments 2. In addition to this, the respected Authors are most affably requested to consider the citing of a single reference throughout occurring (edible plant species, known as a herbal remedy) fatty acid ingredients the text of their enlightening MS related to the diversity of naturally occurring (edible plant species, known as a herbal remedy) fatty acid ingredients; - Natural Product Research 2012, 26(8), 696–702 DOI:10.1080/14786419.2010.550580

Answer 2. We cited the reference, [19] in our manuscript.

Round 2

Reviewer 1 Report

The authors added some references to justify their choice to use TAO for 3 days, as a maximal exposure, for the treatment in HepG2 cells. 

Regarding my suggestion to better investigate the effect of TAO in processes such as immunoactivity, anti-apoptosis, cell proliferation, the authors just answered modifying a sentence in the text.